# Why Do Customers Want to Buy COVID-19 Medicines? Evidence from Thai Citizens

**DOI:** 10.3390/ijerph20065027

**Published:** 2023-03-13

**Authors:** Long Kim, Siwarit Pongsakornrungsilp, Pimlapas Pongsakornrungsilp, Teerasak Jindabot, Vikas Kumar

**Affiliations:** 1Center of Excellence for Tourism Business Management and Creative Economy, Department of Digital Marketing, School of Management, Walailak University, Nakhonsithammarat 80160, Thailand; 2Center of Excellence for Tourism Business and Creative Economy, Department of Tourism and Prochef, School of Management, Walailak University, Nakhonsithammarat 80160, Thailand; 3Faculty of Management Sciences, Prince of Songkla University, Songkhla 90110, Thailand; 4Faculty of Business, Law and Social Sciences, Birmingham City University, Birmingham B25 BCITY, UK; 5Department of Management Studies, Graphic Era Deemed to be University, Dehradun 248002, India

**Keywords:** purchase intention, sales performances, COVID-19 medicines, perceived value, country of origin

## Abstract

Purchase intention has been acknowledged as an important factor influencing businesses’ sales performances and sustainability. Thus, finding factors that influence purchase intention is essential to all relevant businesses. Based on the current importance of purchase intention to businesses, the main objective of this research was to investigate how country of origin, brand image, and perceived value influenced intentions to purchase COVID-19 medicines among Thai consumers. To accomplish this objective, researchers created a Google Form to survey 862 people around Thailand. However, researchers found only 653 valid data, which were analyzed through the structural equation model. The research findings revealed that perceived COVID-19 medicine value increased once the values of country of origin and brand image were highly considered by consumers. At the same time, consumers attempted to buy COVID-19 medicines for their COVID-19 treatments if the products contained high country of origin and perceived values. Finally, the perceived value was found to be a full mediator between brand image and purchase intention. In comparison, despite country of origin and perceived value being the significant factors in purchase intention, consumers’ intention to purchase COVID-19 medicines depended significantly on the degree of the consumers’ perceived value because perceived value contributed the highest impact on purchase intention. These results revealed that COVID-19 medicines were highly valued by many consumers because these medicines could help prevent heavy illness in consumers. As a result, consumers had a higher intention to purchase these medicines for their future COVID-19 treatments.

## 1. Introduction

Understanding consumer behavior can help firms maintain their business sustainability because consumer behavior reveals consumer insights that explain the reasons of their purchase decision [1,2]. In particular, purchase intention, which is one consumer behavior, is a key advantage to many businesses [2]. Recently, many firms and researchers have paid significant attention to consumer purchase intention [3,4,5] because once they are able to deeply understand consumer purchase intention, they can find new solutions to satisfy their target customers and achieve more sales performances in the future [1,2].

Meanwhile, since Thailand experienced two years of the COVID-19 pandemic, 82% of Thai citizens have already received vaccinations, including vaccination boosters [6]. Even though Thailand, these days, has maintained a low rate of COVID-19 infection (around 600 new cases per day) across the country [7], citizens’ social lives and daily business operations are affected by COVID-19 as long as the current pandemic still exists. To help Thai citizens live and cope with the current pandemic even better, the government of Thailand has approved its citizens to use COVID-19 medicines (e.g., Paxlovid for COVID-19 treatment, produced by Pfizer company) recommended by WHO [8]. The current COVID-19 medicines have become important products to Thai citizens because these kinds of medicines can help not only prevent patients from becoming heavily ill but also help patients to recover faster [9] so that they can go back to work normally. Thus, COVID-19 medicines can be highly considered by many Thai citizens. However, reports regarding to Thai citizens’ intention to purchase COVID-19 medicines have not been extensively revealed yet. MBA Skool [10] and Morwitz [2] have underlined that when firms receive enough information about consumers’ intention to purchase their products, they can well manage their supply for current market demands and can obtain high revenues. Based on this scenario, if the relevant medical supply firms that produce COVID-19 medicines can understand Thai consumers’ purchase intention, they can supply sufficient COVID-19 medicine to the current market demand. Thus, investigating the intention to buy COVID-19 medicines among Thai consumers is very important.

Since enhancing consumer purchase intention can help firms promote high sales performances and profits, some researchers have provided suggestions to increase consumer purchase intention in different industries. For instance, Ali et al. [11], from a social commerce context, indicate that country of origin can help promote purchase intention. Products can attract customers’ interest to buy when their country of origin is promoted to the public. In fact, people feel confident using products that are produced in highly developed countries, as these products are considered to have high product quality. On the other hand, Shin and Choi [12] from a hotel service context emphasize brand image as a significant factor in purchase intention. Customers normally think that they can receive good products to service their utility purposes if the products contain a good brand image; thus, they definitely want to buy these products from the firms. In contrast to the above researchers, Zhao and Chen [5], from a green housing context, consider perceived value as the main predictor of purchase intention. Once products offer more benefits compared to their cost, they possibly consider buying them later.

Although brand image, country of origin, and perceived value are respectively considered the predictors of purchase intention in different contexts, their impacts on Thai citizens’ intention to buy COVID-19 medicines have remained unspecified because consumers who are from different contextual studies do not have the same attitudes and perspectives [13]. Meanwhile, the existing literature has not provided enough information to explain how these factors influence purchase intention in the medical product context, particularly in the COVID-19 medicine context. As a consequence, this lack of information has revealed the low awareness of consumer insights regarding the degree of purchase intention in the COVID-19 medicine context. To contribute to the existing literature, this research proposes integrating the above factors into a new theoretical model and aims to investigate (1) how country of origin and brand image affect the perceived value of COVID-19 medicines and (2) how country of origin, brand image, and perceived value affect intention to buy COVID-19 medicines.

## 2. Literature Review

### 2.1. Purchase Intention

Purchase intention refers to a person’s aspirations that cause them to buy a product based on a particular circumstance [1,14]. Once the degree of purchase intention increases, this leads to high purchasing behavior [15]. To evaluate an individual’s behavior, the theory of planned behavior is used to explain how the individual’s behavior develops based on certain reasons [16]. Based on the current concept of the TPB, a person’s perspectives and attitudes are influenced by particular factors that cause them to come up with a decision to take action after assessing resources and chances (e.g., money, skills, ability, time, effort, etc.). Intayos et al. [16] also add that they may possess a higher intention to do a certain thing whenever a person perceives those factors as logical or reasonable causes of actions. In this regard, the TPB has been recognized and used to examine various factors in consumer purchase intention, as this theory is a basic concept that helps researchers to estimate individuals’ buying intention and behavior in specific industries [17]. Therefore, lots of studies have been conducted to investigate consumer purchase intention in various contexts. For example, Aw et al. [18] applied a partial least square–structural equation model (PLS-SEM) to test the impacts of self-congruence, brand prominence, and value perception on purchase intention in a solar product context. In the smartphone industry, Wongsawat and Deebhijarn [19] applied a linear structural relation (LISREL) to test the impacts of marketing mix, preferences, innovative products, and satisfaction on purchase intention. In the online shopping service industry, Ma et al. [20] applied a structural equation model (SEM) to test shopping experience, cognitive involvement, and affective involvement with purchase intention. In the insurance service industry, Nursiana et al. [21] applied the SEM to test perceived risk, product quality, reputation, and service quality with purchase intention. In the tourism industry, Song et al. [22] applied analysis of covariance to test the impacts of perception of competition and social message cues on purchase intention. In the real estate industry, Dash et al. [23] applied the SEM to test the impacts of brand image, integrity, interaction, identity, and satisfaction on purchase intention.

According to scholars’ perspectives from different industries, they respectively outline country of origin [11], brand image [12], and perceived value [5] as potential factors in purchase intention. First, country of origin indicates a specific place from where products originated [24]. It is believed that products are perceived as high quality if those products are produced in developed countries [25]. For instance, products that are produced in Japan or the United States are considered to have a higher quality than local brands [26]. Consumers seem to have high trust in and want to purchase these products for their daily consumption [27]. Therefore, products that have a certain country of origin significantly influence individuals’ interest in purchasing these products.

On the other hand, brand image is considered an essential factor influencing individuals’ purchase decisions [28]. Any brand that has a highly reputable image attracts consumer attention and satisfaction, which eventually creates favorable results and associations in many consumers’ minds over a long period of time [29]. Products with a strong brand image normally have high consumer perception of brand ethics, consistency of advertising, and brand promotion, as well as brand reputation, which further affects individuals’ trust in these products [30]. It is believed that increasing brand image can capture consumers’ needs, wants, and desires, which leads to an enhancement in transactions between consumers and goods and services [23]. Thus, a greater brand image simply influences purchase intention among consumers.

Regarding the perceived value, this displays benefits that consumers obtain from using the products [31]. To check product value, consumers have to assess three main values, namely, social value, emotional value, and functional value [32]. Social value indicates product quality that is derived from a certain group of people [33]. Next, emotional value indicates the degree of pleasure that results from product affection [34]. Last but not least, functional value highlights the degree of product performance that can serve individuals’ needs and wants [35]. Consumers can have strong positive feelings and evaluation toward a certain brand if they are influenced by one of these values.

In the current research gap analysis, although previous investigations considered various factors in purchase intention in different industries, not many researchers have tested the impacts of country of origin, brand image, and perceived value on purchase intention in the medical product industry, especially in the COVID-19 medicine context. As testing these factors has been lacking in the existing literature, this research aims to integrate these antecedents into the purchase intention model and investigate how these antecedents affect consumer purchase intention in the COVID-19 medicine context.

### 2.2. Country of Origin and Perceived Value

Country of origin is conceptualized as consumers’ mental belief triggered by country image that affects their product or service evaluation [11]. In a product assessment, many consumers can judge certain products as inferior or superior products based on where they are produced [26]. In general, many people have a high level of confidence in purchasing products that are produced in developed countries rather than less developed countries [36].

Based on conceptualized comparisons, a good country of origin can increase consumer confidence in using the products with the firms [11], while high perceived product value attracts more consumer interest to use the products with the firms [37]. The above conceptual comparisons outline positive views between country of origin and perceived value. In brand equity behavior, many people expect more benefits from products that are produced in highly advanced countries [38]. Based on a review of consumer perception, consumers strongly value products that are originally from developed countries [39]. According to these theoretical explanations, the link between country of origin and perceived value seems to be positive. In the vehicle industry, Abdelkader [40] found that country of origin has a significant effect on perceived value. In the social commerce industry, Ali et al. [11] underscore that country of origin positively affects perceived value. In this regard, we hypothesize the current relationship in H1.

### 2.3. Country of Origin and Purchase Intention

A strong country of origin positively influences individuals’ product evaluations [41], while a high individual purchase intention develops a positive decision to purchase products [26]. These variables underline positive directions. In car consumer behavior, a strong propensity of desire to buy products happens among consumers when the products are from highly advanced countries [42]. From a social commerce perspective, consumers demonstrate positive buying attitudes toward a brand once its country of origin is positively perceived by those consumers [11]. Based on the above explanations, the link between country of origin and purchase intention is positive. According to the e-commerce industry, Nurunnisha et al. [43] reveal that country of origin shows a positive influence on purchase intention. In the smartphone industry, Prahiawan et al. [36] consider country of origin as a positive factor in purchase intention. In this regard, we hypothesize the current relationship in H2.

### 2.4. Brand Image and Perceived Value

Brand image refers to customer perceptions about a certain brand name, which has a connection with those consumers’ memories [44]. Strong brand image can develop a strong link (between the users and brands), which leads to brand association [45]. In brand association, there are three main classifications (attributes, benefits, and attitudes) that demonstrate brand strengths, favorability, and uniqueness to consumers [44,45].

Strong brand image can increase individuals’ favorable desire toward products [45], while perceived value indicates individuals’ positive perspectives on a certain brand [37]. These variables’ concepts raise similar directions. In smartphone consumer behavior, consumers perceive more benefits from using products from a certain brand when they strongly acknowledge the brand image [46]. Based on a study from retail store perspectives, good brand image can gain more trust from customers; therefore, its product value is considered high by many users [47]. Based on these discussions, brand image and perceived value have a positive connection. In online travel agencies, Pham and Nguyen [48] demonstrate that brand image can positively influence perceived value. In the dental clinic industry, Lin and Yin [46] reveal that brand image is a positive predictor of perceived value. In this regard, we hypothesize the current relationship in H3.

### 2.5. Brand Image and Purchase Intention

Developing a good brand image can make consumers have more positive perceptions of its products [12], whereas high purchase intention indicates a strong aspiration to buy products from companies [49]. These two concepts exhibit positive views. In fashion purchasing attitudes, many consumers feel confident with a good brand image, which causes them to have a high desire to buy products from firms [29]. In Facebook purchase behavior, many consumers feel safe investing their money in a particular brand that has a good image [50]. Based on the above discussions, brand image and purchase intention show a positive connection. In the smartphone industry, Savitri et al. [51] reveal that a strong brand image increases individuals’ purchase intention. In the fashion industry, Chen et al. [29] highlight a positive link between brand image and purchase intention. In this regard, we hypothesize the current relationship in H4.

### 2.6. Perceived Value and Purchase Intention

Perceived value refers to an overall product assessment that is an outcome of a comparison between sacrifices and benefits [31]. Based on the means–end theory, an idea of value is derived from a direct comparison between price and product quality [52]. Kim et al. [37] reveal that a product that is considered to be valuable needs to have three main values: emotional value, functional value, and social value.

A high perceived value indicates positive attitudes among consumers [53], while strong purchase intention develops an actual purchase behavior among consumers [54]. These two variables highlight similar directions. Based on purchase behavior in social commerce, when the products are considered to offer more benefits to users, they can increase users’ desires to purchase [55]. According to purchase behavior in eco-friendly athleisure apparel, once consumers strongly perceive the product value, they definitely want to purchase the product from the firm [56]. The above discussions reveal that perceived value is most likely to have a positive relationship with purchase intention. In the online shopping service industry, Yin and Qiu [57] emphasize perceived value as a positive factor in purchase intention. In the greenhouse industry, Zhao and Chen [5] mention that high perceived value increases individuals’ purchase intention. In this regard, we hypothesize the current relationship in H5.

### 2.7. Aims and Hypotheses

According to the contemporary literature, this paper attempts to fill in the current gap in research by developing a theoretical model that can be applied to investigate factors in purchase intention for medical products, especially in COVID-19 medicines. Thus, the main objective of this research paper is to examine how country of origin, brand image, and perceived value affect the intention to purchase COVID-19 medicine among Thai consumers. According to the above theoretical discussions, the systematic impacts among variables (brand image, country of origin, perceived value, and purchase intention) are developed in this research paper. Therefore, all the hypotheses are formulated below (Figure 1):

**H1.** 
*Strong country of origin gains consumers’ perceived COVID-19 medicine value.*


**H2.** 
*Strong country of origin increases consumers’ intention to purchase COVID-19 medicines.*


**H3.** 
*Good brand image gains more consumers’ perceived COVID-19 medicine value.*


**H4.** 
*Good brand image increases consumers’ intention to purchase COVID-19 medicines.*


**H5.** 
*High perceived value increases consumers’ intention to purchase COVID-19 medicines.*


## 3. Methods

### 3.1. Sample and Data Collection

The main objective of this research was to investigate the intention to purchase COVID-19 medicines in Thailand. In order to be qualified participants in the survey processes, all participants needed to be at least 18 years old. At the same time, this research was investigating participants’ intention attitudes toward purchase rather than purchase behavior; thus, the most suitable participants of this study were the ones who had never bought COVID-19 medicines before. On the other hand, if they had already bought COVID-19 medicines, they were not eligible for this study. Based on the current sample criteria, all of the participants who were invited must have never experienced buying COVID-19 medicines at any pharmacy around Thailand. In this regard, 862 people who were living in different areas of Thailand were invited to join the survey. In addition, researchers created a Google Survey form to allow all interested participants to fill in the survey.

In data collection, a convenience sampling technique was employed to survey opinions from all qualified participants. First, researchers contacted these people through social media platforms such as Facebook, Line, and Instagram. Once they were found, they were given a screening question “Have you ever bought any COVID-19 medicine before?” After they provided a “Yes” answer, they were asked to fill in self-administered questionnaires. Next, when they agreed to join, they were given a Google Survey link. Finally, all of the data were fully gathered with a 100% response rate. However, only 76% of the valid response rate (653 valid data) were kept for data analysis after passing a careful cleansing process.

### 3.2. Survey Construct

A survey construct consists of four main variables. Researchers originally borrowed items of each variable from previous research. First, three items of country of origin were borrowed from Ali et al. [11]. Second, three items of brand image were borrowed from Rivai and Zulfitri [58]. Third, three items of perceived value were borrowed from Kim et al. [37]. Last but not least, three items of purchase intention were borrowed from Liu et al. [55]. Therefore, all of the questionnaire items were constructed in Appendix A Table A1. 

In order to obtain answers from the participants, the researchers employed a 5-point Likert scale to let participants rate their opinions on each item. This kind of Likert scale was considered an easy method to answer all questionnaire items because it had a mid-scale (3 = neutral), which created a clear line between positive and negative options [53]. Kim et al. [59] also supported that using this technique could be a reasonable technique, as the current rating technique did not consume lots of time and effort from participants to complete the whole survey.

### 3.3. Data Analysis Technique

The valid data of this study were analyzed using the structural equation model (SEM). In the SEM technique, researchers followed Anderson and Gerbing’s two-step approach [60]. The current two-step approach contained both the measurement and structural models, which were estimated using statistical software with a maximum likelihood (ML) estimation [61]. First, the structural model was constructed using confirmatory factor analysis (CFA) to assess model fitness to perform regressions among variable relationships, ensuring all fitness indicators (e.g., CMIN^2^/df, GFI, NFI, CFI, AGFI, RMSEA, and PCLOSE) pass the fitness minimum requirements [37]. Next, model measurements were estimated to find construct reliability, convergent validity, and discriminant validity [62]. Once these steps were completed, performing regressions using the structural equation model technique was finally conducted. In this regard, standardized coefficient betas between relationships could be obtained; hence, these results could help researchers to justify all hypotheses in this study.

## 4. Research Findings

### 4.1. Model Construct

Researchers analyzed all valid data using a structural equation model (SEM). However, researchers needed to present model fitness and the measurement model in the SEM technique. First, model fitness was modified by applying a confirmatory factor analysis to push fitness indicators (CMIN^2^/df, GFI, NFI, CFI, AGFI, RMSEA, and PCLOSE) to reach the minimum thresholds of fitness recommended by Kim et al. [37]. According to Table 1, the model was fit, as all fitness indicators already passed the thresholds after fitness modification.

Finally, the variable measurements were assessed to ensure loading factors, content reliability, and convergent validity in Table 2. First, the loading factor scores were checked and kept if any item obtained scores above 0.6. Second, content reliability was checked using Cronbach’s alpha and composite reliability scores (CRs) (scores > 0.7) [59,63,64]. In Table 2, both Cronbach’s alpha and CR scores are bigger than 0.7, indicating content reliability. Finally, the convergent validity was checked and passed the requirement scores of average variance extracted (AVE) (scores > 0.5), indicating the convergent validity [62,65].

### 4.2. Results of Structural Equation Model

All main results and other critical ratios are briefly recorded in Figure 2 and Table 3. Regarding the impacts on perceived value, country of origin significantly influenced perceived value (β = 0.60, *p* < 0.001), which supports Hypothesis 1. Furthermore, brand image significantly influenced perceived value (β = 0.40, *p* < 0.001), which supports Hypothesis 3.

Regarding the impacts on purchase intention, perceived value significantly influenced purchase intention (β = 0.61, *p* < 0.001), which supports Hypothesis 5. Next, country of origin significantly influenced purchase intention (β = 0.40, *p* < 0.001), which supports Hypothesis 2. In contrast, purchase intention was not significantly influenced by brand image (β = 0.01, *p* > 0.05), which does not support Hypothesis 4.

Regarding to the mediation test of brand image → perceived value → purchase intention, the indirect impact of brand image on purchase intention was significant (β = 0.27, *p* < 0.05), while its direct impact on purchase intention was insignificant (β = 0. 01, *p* > 0.05). Therefore, perceived value became a full mediator between brand image and purchase intention. Based on these results, the hypotheses are finally reported in Table 3.

## 5. Discussion

Regarding to the impacts on perceived value, country of origin positively affected perceived value. Strong country of origin revealed strong product quality, which was normally perceived by many consumers [11]. Therefore, this situation indicates that the products that had a strong country of origin were considered to provide high performances. Thus, this further triggered public opinions to highly value the products. In fact, the current results were similar to previous studies [11,40], which revealed that consumers perceived a high value when the products were produced in advanced countries. Based on this result, Thai consumers seem to highly value COVID-19 medicines that are produced in developed countries (e.g., the US or Japan). Next, brand image positively affected perceived value. A better image indicated high product quality, which attracted public trust [46]. Once the products contained a strong brand image, these products gained positive views among consumers who possibly saw the products’ values or benefits for their consumption. These results display similar findings to previous research [46,48], which indicated that consumers positively estimated product benefits for their consumption after acknowledging a strong brand image on the product packages. In this case, if COVID-19 medicines had strong brand images, the consumers were likely to positively evaluate the current medical products for their future COVID-19 treatments.

Regarding the impacts on purchase intention, perceived value positively affected purchase intention. Product value showed a certain degree of product performance or benefit, which was expected to provide benefits to consumers [57]. This situation underscores a positive assessment from consumers who, in turn, had more desire to buy the products. Similarly, previous studies also agreed that consumers really wanted to buy products after they perceived the benefits of the products [5,57]. Based on this scenario, when COVID-19 medicines attracted more positive evaluations from the consumers, this also attracted Thai consumers to consider buying the current medical products for their future COVID-19 treatments. Next, country of origin positively affected purchase intention. Any brand that originated in advanced countries was generally considered to deliver high product performance to consumers [36]. From a psychological perspective, among consumers, products from advanced countries usually qualified for and passed the international quality requirement or standard. Therefore, people strongly believed in and trusted the products, which led to a high willingness to pay for these products. Obviously, these results are also similar to previous investigations [36,43], which highlighted that products produced in advanced countries could strongly convince consumers’ purchase intention. This circumstance reveals that the brands of COVID-19 medicines that have a better country of origin could definitely increase Thai citizens’ purchase intention. In contrast, brand image did not show any effect on purchase intention. Its insignificant relationship with purchase intention happened because perceived value stood as a full mediator between brand image and purchase intention. Therefore, this caused brand image to demonstrate an indirect impact on purchase intention, which marked an opposite result to previous studies [29,50,51]. This result indicates that individuals’ intention to purchase COVID-19 medicines from a certain brand name depended significantly upon the product value, which was highly supported by the influence of brand image. People gave more value to COVID-19 medicines which ultimately triggered their desire to buy the medicines for their future COVID-19 treatments. In the middle of the COVID-19 pandemic, people first thought of having medicines to treat their sickness or save their lives, while brand image was not the primary concern among consumers. Therefore, the side effect of brand image did not influence consumers’ intention to purchase COVID-19 medicines.

## 6. Theoretical and Managerial Implications

In light of this research, our research provides significant contributions to the existing literature. First, this research extends our understanding of purchase intention by integrating country of origin, brand image, and perceived value into the theoretical model of purchase intention in the medical product industry, particularly COVID-19 medicine products. Previous studies have considered country of origin [36,43], brand image [29,51], and perceived value [5,57] as the main predictors of purchase intention in their contextual industries, respectively. However, the existing literature has not provided enough information on how these factors influence consumers’ intention to purchase COVID-19 medicines. Therefore, we included these factors in the theoretical model of intention to purchase COVID-19 medicines to extend our understanding of how consumers attempt to buy COVID-19 medicines for their COVID-19 treatments, following their evaluation of the influences of the above factors (e.g., country of origin, brand image, and perceived value) on their purchase intention. Furthermore, our current theoretical model of intention to purchase COVID-19 medicines outlines the full mediating effect of perceived value on the relationship between brand image and purchase intention. This finding causes brand image to have an indirect impact on purchase intention, which is different from previous studies that considered the direct influences of brand image on purchase intention in their industries [29,50,51]. The current finding has deeply increased our awareness of how brand image significantly enhances the value of products, which ultimately influences consumers’ attempts to buy COVID-19 medicines for their COVID-19 treatments. Therefore, we acknowledge the significant role of perceived value in promoting individuals’ intention to purchase COVID-19 medicines.

From a managerial perspective, promoting perceived COVID-19 medicine value and products’ country of origin to consumers can lead to high purchase intention. First, marketers can promote perceived COVID-19 medicine value by crafting product quality to ensure that consumers use it effectively to cure their COVID-19 disease. This can help reduce the mortality rate of consumers as well as help them recover faster. At the same time, products are required to have high packing safety. This can help products avoid being damaged while they are shipped or transported to markets. Therefore, consumers have a high level of confidence in purchasing products from firms or pharmacies. Second, marketers can promote perceived COVID-19 medicine value using brand image. Appropriate marketing needs to be conducted to spread more information to consumers. Marketers may use various tools, such as Facebook, YouTube, Twitter, events, and other advertisements, to promote and increase brand awareness among their consumers. Last but not least, the products’ country of origin has to be mentioned on the products. In particular, Thai citizens have higher intentions to purchase products that are from advanced countries, such as the United States or Japan, as previous studies also support that products produced in these countries are considered to have high product quality [26]. Therefore, the information “made from the US or Japan” has to be mentioned on product packaging so that it can further attract individuals’ trust, which results in a higher intention to buy these products for their consumption.

## 7. Research Limitations and Future Studies

Despite accomplishing the main objective of this research, some limitations appeared in this research. First, the results of this research were mainly from Thai consumer perspectives; thus, there could be some bias among answers. Future research should be conducted in other countries to understand the perspectives of other people regarding their intention to buy COVID-19 medicines. Second, the results of this research could be difficult to generalize to other contexts such as cosmetic, green car, and other e-service industries. Future research can adopt these factors to further investigate those contexts to come up with new findings and conclusions. Finally, the participants in this research were able to fill out the Google survey form by themselves, and the researchers were not able to control and observe their answers. Thus, this may have also caused some bias among answers. Future research should use the convenience sampling method to directly ask participants to fill out the survey at convenient places so that they can, at least, control the survey processes.

## 8. Conclusions

The main objective of this research was to investigate the impacts of country of origin, brand image, and perceived value on the intention to purchase COVID-19 medicines among Thai consumers. To accomplish this objective, researchers developed a Google survey form to survey 862 people around Thailand. After that, researchers applied the SEM technique to analyze 653 valid data. The research findings indicate that the value of COVID-19 medicines was perceived as high when consumers saw a strong country of origin and brand image of the products. Furthermore, consumers had higher intentions to buy COVID-19 medicines once they acknowledged the advanced country of origin (e.g., the United States or Japan) on the product packages. Last but not least, consumers wanted to buy COVID-19 medicines after they saw the value of COVID-19 medicines (e.g., reduced mortality rate, fast recovery, and safety of usage) for their consumption.

## Figures and Tables

**Figure 1 ijerph-20-05027-f001:**
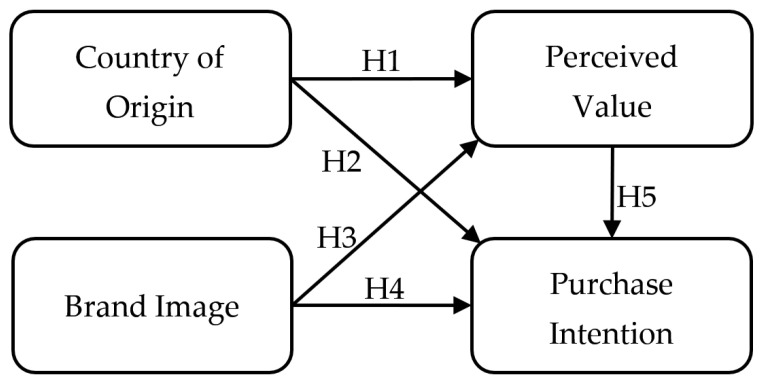
A Theoretical Model of Purchase Intention.

**Figure 2 ijerph-20-05027-f002:**
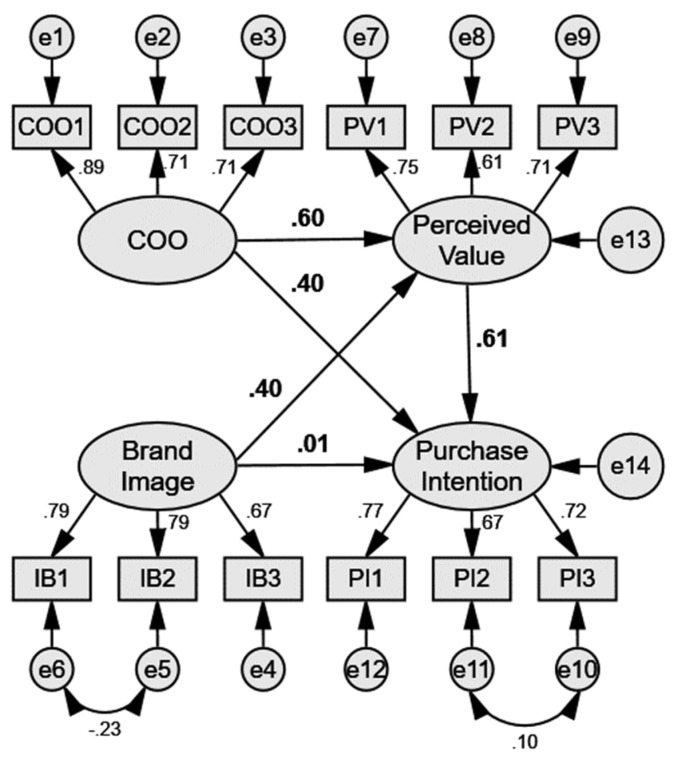
SEM Results.

**Table 1 ijerph-20-05027-t001:** Model Fitness.

Indicator	Index		Thresholds	Results
	Before Modification	After Modification		
CMIN^2^/df	1.401	1.107	≤3	Good
GFI	0.905	0.956	>0.9	Good
NFI	0.884	0.958	>0.9	Good
CFI	0.968	0.978	>0.9	Good
AGFI	0.879	0.993	>0.8	Good
RMSEA	0.078	0.041	<0.08	Good
PCLOSE	0.768	0.818	>0.05	Good

**Table 2 ijerph-20-05027-t002:** Measurement Model.

Variable	Items	Loading Factor	Cron. Alpha	CR	AVE
Country of Origin	COO1	0.89	0.88	0.70	0.76
	COO2	0.71			
	COO3	0.71			
Brand Image	BI1	0.79	0.92	0.77	0.81
	BI2	0.79			
	BI3	0.67			
Perceived Value	PV1	0.75	0.74	0.78	0.79
	PV2	0.61			
	PV3	0.71			
Purchase Intention	PI1	0.77	0.73	0.82	0.74
	PI2	0.67			
	PI3	0.72			

**Table 3 ijerph-20-05027-t003:** Results and Hypotheses Summary.

**Panel A: Regressions and Critical Ratios**						
** *Hyp.* ** ** *No.* **	** *Proposed Relationships* **		** *Std. Beta (β)* **	** *p-Value* **	** *Sig.* ** ** *Level* **	** *Hyp.* ** ** *Result* **
	** *Independent Variable* **	** *Dependent Variable* **				
1	Country of Origin	Perceived Value	0.60	0.000 **	Sig.	Supported
2	Country of Origin	Purchase Intention	0.40	0.000 **	Sig.	Supported
3	Brand Image	Perceived Value	0.40	0.000 **	Sig.	Supported
4	Brand Image	Purchase Intention	0.01	0.228	Insig.	Not Supported
5	Brand Trust	Purchase Intention	0.61	0.000 **	Sig.	Supported
**Panel B: Mediation Testing**						
** *Relationships* **			** *Indirect* **	** *Direct* **	** *Mediation* **	** *Result* **
Brand Image → Perceived Value → Purchase Intention			0.27 *	0.01	Full Mediation	Sig.

Note: * indicates sig. level of *p* < 0.05, whereas ** indicates sig. level of *p* < 0.001.

## Data Availability

Data are not publicly available due to high privacy restriction.

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
