# Peer review of "Why Do Customers Want to Buy COVID-19 Medicines? Evidence from Thai Citizens"

_ijerph, 2023, doi:10.3390/ijerph20065027_

Round 1
Reviewer 1 Report
Dear Authors,
I congratulate you on an interesting study and a well-crafted text. The only objections I have after reading it relate to the references to the literature in the text - in several places the name of the author(s) and year of publication are given - e.g. MBA Skool (2021) and Morwitz (2014), line 50, while in my opinion it should be: MBA Skool [10] and Morwitz [2].
I would also suggest a separate subsection (#7) outlining the limitations of the study, currently found a Conclusion (rather Conslusions) section.
Author Response
Point 1: The only objections I have after reading it relate to the references to the literature in the text - in several places the name of the author(s) and year of publication are given - e.g. MBA Skool (2021) and Morwitz (2014), line 50, while in my opinion it should be: MBA Skool [10] and Morwitz [2].
Response 1: We already correct the citation errors through the paper. For instance, information can be found at Line: 55, Line:64, Line:69, Line: 73, etc.
Point 2: I would also suggest a separate subsection (#7) outlining the limitations of the study, currently found a Conclusion (rather Conclusions) section.
Response 2: Researchers created a new section “ 7. Research Limitations and Future Studies” and included all relevant information. Information can be found in Line: 428-440.

Reviewer 2 Report
This is an interesting study written about how the country of origin, brand image, and perceived value influenced purchase intentions for COVID-19 medicines.
However, there are some parts that need to be reinforced on these points:
1. The most important supplement is necessary to explain the definition in detail and how to measure the variables 'strong country of origin' and 'good brand' and the terms. In other words, it needs to explain in detail the measurement of main variables, such as which brand is good and which brand is strong, rather than strong and good as average values. For example, the results of this study also differ depending on whether the brand is a fictitious brand or a brand already known to consumers. It is necessary to explain how to measure in detail.
2. Also, it is necessary to supplement the literature review and how to measure the variables by attaching main papers such as the country of origin effect and brand image among previous studies.
3. It needs to explain the contribution of practicals and how to use the application of the results in detail through demonstrate the relationship between the COO and PI. So it is necessary to enhance the content of the conclusion.
Author Response
Point 1: The most important supplement is necessary to explain: definition in detail and how to measure the variables 'strong country of origin' and 'good brand' and the terms. In other words, it needs to explain in detail the measurement of main variables, such as which brand is good and which brand is strong, rather than strong and good as average values. For example, the results of this study also differ depending on whether the brand is a fictitious brand or a brand already known to consumers. It is necessary to explain how to measure in detail.
Response 1: We already provided extra information to further justify the measurement of strong COO and strong brand. Information can be found at:
(1) for COO, at Line: 121-122
(2) for brand, at Line: 129-132.
Point 2: Also, it is necessary to supplement the literature review and how to measure the variables by attaching main papers such as the country-of-origin effect and brand image among previous studies.
Response 2: We already provided extra information to further explain the effects of these variables in the literature review. Information can be found at Page: 3, Line:117-143.
Point 3: It needs to explain the contribution of practical and how to use the application of the results in detail through demonstrate the relationship between the COO and PI. So it is necessary to enhance the content of the conclusion.
Response 3: We already provided more explanation regardless of relationship between COO and PI. At the same, content of the conclusion was also enhanced. Information can be seen at “In managerial perspectives” Line: 421-427.

Reviewer 3 Report
The authors present a well-written and compelling paper on the factors that influence purchase intention in the context of COVID-19 in Thailand, specifically examining the impact of country of origin, perceived value, and brand image. The paper is clearly motivated and addresses an important topic that will be of interest to scholars and practitioners alike. Overall, I have only minor concerns about the paper.
References: The paper contains both APA and IEEE references. It would be helpful to ensure consistency in the referencing style throughout the paper. For example, please check lines 50, 59, 64, and 98 for inconsistencies.
Facebook: The authors should ensure consistency in spelling the name of the social media platform. Please check lines 175 and 229 to ensure that it is spelled consistently.
COO: The authors introduce the concept of "country of origin" (COO) in line 115 but do not use the term consistently until the results section. It would be helpful to use the full term "country of origin" throughout the paper for clarity.
Questionnaire: It would be helpful to include the questionnaire in an appendix or provide more detail on the questions in the methodology section to aid in reproducibility.
Relationship between BI and PI: It would be useful to explore the reasons for the poor relationship between "brand image" and "purchase intention" in the study.
Abstract: The last sentence of the abstract could be clearer. It would be helpful to revise this sentence for greater clarity.
Theoretical justification: It would be helpful to include more paragraphs in the theory section that provide justification for the hypotheses based on previous studies with inconclusive findings regarding the relationships between the variables.
Author Response
Point 1: References: The paper contains both APA and IEEE references. It would be helpful to ensure consistency in the referencing style throughout the paper. For example, please check lines 50, 59, 64, and 98 for inconsistencies.
Response 1: We already changed the references throughout the paper. Reviewer can these consistencies, e.g., line: 64, Line:69, Line: 73, Line: 98, etc.
Point 2: Facebook: The authors should ensure consistency in spelling the name of the social media platform. Please check lines 175 and 229 to ensure that it is spelled consistently.
Response 2: We already corrected the spelling of the name of social media platform. Answers can be found at line: 206 and Line: 266.
Point 3: COO: The authors introduce the concept of "country of origin" (COO) in line 115 but do not use the term consistently until the results section. It would be helpful to use the full term "country of origin" throughout the paper for clarity.
Response 3: We already changed and wrote only “country of origin” throughout the manuscript following reviewer suggestion. Information can be found at Line: 146, Line: 322, Line:328, Line: 338, etc.
Point 4: Questionnaire: It would be helpful to include the questionnaire in an appendix or provide more detail on the questions in the methodology section to aid in reproducibility.
Response 4: We are already move questionnaire items to Appendix. Information can be found at page:11, Line:464.
Point 5: Relationship between BI and PI: It would be useful to explore the reasons for the poor relationship between "brand image" and "purchase intention" in the study.
Response 5: We already provided further explanation the insignificant relationship between BI and PI. Information can be found at Page: 9, Line: 382-385.
Point 6: Abstract: The last sentence of the abstract could be clearer. It would be helpful to revise this sentence for greater clarity.
Response 6: We already further provided explanation to clarify the last sentence of the abstract. Information can be found on page 1, Line: 27-30.
Point 7: Theoretical justification: It would be helpful to include more paragraphs in the theory section that provide justification for the hypotheses based on previous studies with inconclusive findings regarding the relationships between the variables.
Response 7: We already provided extra information in theory section to deepen the connections between variables and purchase intention. Information can be found at Line: 117-143.

Reviewer 4 Report
Please provide the following improvements:
1. representativeness and validity of sample - this needs to be elaborated
2. necessary to show used methods and materials for analysis
3. please provide some references from the Journal where you aspire to publish the paper
4. please further elaborate the discussion and conclusions
Author Response
Point 1: representativeness and validity of sample - this needs to be elaborated
Response 1: We have added more information to further explain representativeness and validity of sample study: Information can be found at Line: 254—260, Line: 264-268.
Point 2: necessary to show used methods and materials for analysis
Response 2: We already created new section to further explain the method and material for analysis of this study. Information can be found at Line:286-299.
Point 3: please provide some references from the Journal where you aspire to publish the paper
Response 3: We already cited 3 papers as references from journal where I aspire to publish the paper. Information can be seen at Line: 475 , Line: 494, and Line: 595.
Point 4: please further elaborate the discussion and conclusions
Response 4: We already provided extra information in discussion and conclusions. Information can be found at:
(1) Discussion: Line: 343-345, Line:350-353, Line: 360-361, Line:369-371, Line: 382-385.
(2) Conclusion: Line: 446-452.

Round 2
Reviewer 2 Report
It is accepted as much as the part that needs to be supplemented in the research contents is mentioned.
Reviewer 4 Report
Ok